# Development of Narrow-Bore C18 Column for Fast Separation of Peptides and Proteins in High-Performance Liquid Chromatography

**DOI:** 10.3390/polym14132576

**Published:** 2022-06-25

**Authors:** Ashraf Ali, Sarah Alharthi, Nora Hamad Al-Shaalan, Eman Y. Santali

**Affiliations:** 1Department of Chemistry, Faculty of Natural Sciences, University of Haripur, Haripur 22062, Khyber Pakhtunkhwa, Pakistan; 2Department of Chemistry, College of Science, Taif University, Taif 21944, Saudi Arabia; sarah.alharthi@tu.edu.sa; 3Department of Chemistry, College of Science, Princess Nourah bint Abdulrahman University, Riyadh 11671, Saudi Arabia; nhalshaalan@pnu.edu.sa; 4Department of Pharmaceutical Chemistry, College of Pharmacy, Taif University, Taif 21944, Saudi Arabia; eysantali@tu.edu.sa

**Keywords:** protein separation, HPLC column, fast analysis, separation efficiency, peptide separation

## Abstract

Separation with high efficiency and good resolution is constantly in demand in the pharmaceutical industry. The fast and efficient separation of complex samples such as peptides and proteins is a challenging task. To achieve high efficiency with good resolution, chromatographers are moving towards small particles packed into narrow-bore columns. Silica monolith particles (sub-2 µm) were derivatized with chlorodimethyl octadecyl silane (C18) and packed into stainless steel columns (100 mm × 1.8 mm i.d) by a slurry-packing method. The developed columns were used for the separation of peptides and proteins. A separation efficiency (N) of 40,000 plates/column (400,000 plates/m) was achieved for the mixture of five peptides. Similarly, the fast separation of the peptides was carried out using a high flow rate, and the separation of the five peptides was achieved in one minute with high efficiency (N ≅ 240,000 plates/m). The limit of detection (DL) and the limit of quantification (QL) for each analyte were determined by developing a linear regression curve with relatively very low concentrations of the target compound. The average values of the QL for the peptide and proteins were 0.55 ng and 0.48 ng, respectively, using short C18 column (1.8 mm × 100 mm) UV (at 214 nm). The fast analysis of peptides and proteins with such high efficiency and good resolution has not been reported in the literature yet. Owing to high efficiency, these home-made columns could be used as an alternative to the expensive commercial columns for peptide and protein separation.

## 1. Introduction

Innovation in HPLC column development is continually progressing, and the requests for high efficiency and fast analysis increase day by day. Owing to the increasing numbers of samples, pharmaceutical industries are looking for rapid analysis with high efficiency [1,2]. The advances in particles, stationary phases, columns, and instrument design have made the ultrafast separations easy, which can reduce the run times by 70% or more. High-efficiency stationary phases and columns are required for the fast analysis of various analytes. Silica particles in the range of 1–2 µm are used for packing the HPLC columns for fast analysis, as compared to the HPLC columns available on the market [3,4,5].

The improvement in chromatographic performance can be realized either in a faster analysis with the same resolution, higher plate number, and better resolution or by some mixture of both benefits. The prospect of time savings and the associated increase in productivity have created tremendous excitement and have raised performance expectations among HPLC users [6,7,8,9,10]. Narrow-bore short HPLC columns (1–2.1 mm id × 3–5 cm) became very popular for the past analysis of biomolecules, especially in the pharmaceutical industry, as they reduce the analysis time, and many samples can be analyzed within a short period of time. In the recent past, the choice of HPLC column dimensions shifted from 4.6 mm (internal diameter) to 2.1 mm or even 1 mm for achieving high-speed high separation efficiency [10]. The development of small particles as HPLC packing materials (sub-2 m PPP and core–shell particles) have made it possible to reduce the column dimensions, and high-speed separations can be achieved [11,12,13,14]. Short narrow-bore HPLC columns have several advantages, such as high sensitivity, low cost, and a less mobile phase consumption, and they require a small amount of sample [15,16,17,18,19]. However, there are two main challenges in using narrow-bore short HPLC columns: the effect of frictional heating at high pressure [20], which reduces column efficiency significantly [21], and particle fracture when using a very high column-packing pressure and challenging column packing, which need a sophisticated column-packing assembly [22]. These challenges should be addressed to achieve the desired efficiency and resolution in fast analysis.

The main purpose of the current work is the development of short narrow-bore columns for the fast analysis of peptides and proteins. There are several parameters which affect column performance, such as the stationary phase (particle size, pore size, surface area, and bonding chemistry), column packing, which needs good skills for the packing of a void-free column, and method development for the separation of targeted analytes (mobile phase selection, mode of separation, flow rate, detection, etc.). An improvement in any parameter may change the results of the separation significantly. The main purpose of the current study is to develop a short narrow-bore column for the fast analysis of peptides and proteins. In this study, sub-2 µm silica monolith (SM) particles were prepared by a sol-gel method using tri-methoxy orthosilicate (TMOS) as a silica source, poly-ethylene glycol (PEG) as a surfactant, and urea as a porogen. The same method was used for the preparation of SM particles as that reported in Refs. [23,24,25,26,27], with few changes in the reaction formulation. These particles were derivatized with chlorodimethyl octadecyl silane (C18) ligand to prepare a reversed phase (RP) stationary phase for the peptide and protein separation. The SP was packed in a narrow-bore glass-lined stainless steel column (100 mm × 1.8 mm) by the slurry-packing method and checked for the separation of a mixture of five peptides (T-T-S; V-A-P-G; AT-I; Isotocine; and Bradykinin) and five proteins, Cytochrome c; RNase A; Lysine; α-Amylase; and Insulin. It was observed that the five peptides were separated within one minute, while all five of the proteins mentioned above were separated within two minutes, using a C18-bound SM column (100 mm × 1.8 mm). The issue of narrow-bore column packing was successfully addressed by using mechanical vibration during the column packing, which improved the packing quality. In the literature, there are several articles on fast analysis, but the analytes are either small molecules (alkyl benzene, etc.) [28,29,30] or peptides [31], and the separation efficiency and resolution are very poor as compared to this study. The efficiency achieved for the peptide and protein separation under the RP-LC mode in the current study is the highest, and such high efficiency has not been reported in the literature so far. So, the short narrow-bore columns (100 mm × 1.8 mm ID) packed with the C18-bonded sub-2 µm silica monolith particles developed in this study could be used for the fast analysis of peptides and proteins.

## 2. Materials and Methods

### 2.1. Chemicals and Apparatus

Polyethylene glycol (PEG, CAS 25322-68-3), acetic acid (CAS 64-19-7), tetramethyl orthosilicate (TMOS, CAS 681-84-5), urea (CAS 57-13-6, hexamethyldisilazane (HMDS) CAS 999-97-3, chlorotrimethylsilane (TMCS) CAS 75-77-4, chlorodimethyloctadecylsilane CAS 18643-08-8, acetonitrile (CAS 75-05-8), acetone (CAS 67-64-1, water CAS 7732-18-5, methanol (CAS 67-56-1), and toluene (CAS 108-88-3) were bought from Sigma–Aldrich (St. Louis, MO, USA). All the solvents used in this study were HPLC grade and used as supplied by the vendor without further purification. The screen frits and glass-lined steel tubing were purchased from Valco (Houston, TX, USA).

### 2.2. Preparation of Stationary Phase for Peptide and Protein Separation

#### 2.2.1. Preparation of SM Particles

The silica monolith particles were prepared in the same way as reported in Ref. [32], with some modification in the heating timing and the ratio of components in the reaction mixture. A mixture of PEG (2 g), urea (2 g), 100 mL 0.01 N acetic acid, and 6 mL TMOS was stirred for 30 min at 0 °C, followed by heating at 40 °C for 12 h in a Teflon container and then heating at 120 °C for 12 h. The extra liquid was decanted off and the product was dried at 70 °C for 10 h. After drying, the product was calcined at 550 °C for 10 h.

#### 2.2.2. C18 Modification of SM Particles

Two grams of calcined SM particles was dispersed in 100 mL anhydrous toluene and stirred for 10 min. Chlorodimethyl octadecyl silane (20 mg) was added into it. The contents were refluxed at 100 °C for 12 h in a round-bottomed flask, followed by the end-capping of the remaining silanol with hexamethyl disilazane (HMDS) and chlorotrimethylsilane (TMCS). After the completion of the reaction, the product was washed, filtered, and dried at 80 °C for 5 h.

### 2.3. Characterization

S-4200 field emission scanning electron microscope (FE-SEM, HITACHI, Hitachi, Japan) was used for the surface analysis of the bare and C18-bonded SM particles. The particle size was measured with a Mastersizer 2000 particle-size analyzer.

### 2.4. Packing HPLC Column

A scree frit (pore size 0.5 µm) was placed inside an empty stainless steel tube (100 × 1.8 mm), and a column was connected to the slurry packer (Altech slurry packer). One hundred and fifty milligrams of SP (C18-bound SM particles) was dispersed in methanol, and the slurry was fed into an empty column attached to the slurry packer. The columns were packed by applying 12,000 psi of pressure. To check the reproducibility, three columns were packed, and their performances were checked using the same elution condition and the same test analytes, and the reproducibility was successfully achieved.

### 2.5. Checking the Performance of Packed Column

A liquid chromatography system consisting of a pump, a detector, and a recorder, as mentioned in Ref. [27], was used for checking the separation performance of the packed column.

A µLC system was constructed by assembling a 10 AD pump (Schimadzu, Tokyo, Japan), an injector (C14 W.05) with a 50 nL injection loop from Valco (Houston, TX, USA), a membrane degasser (Schimadzu DGU-14A), and a UV-VIS capillary window detector (Jasco UV-2075). A glass-lined micro-column (1.8 mm × 100 mm), packed with C18-bonded SM was connected and was installed with a capillary (50 μm I.D. 365 μm O.D.) using a PEEK sleeve, and it was connected to the capillary window detector. The mixtures of peptides and proteins were analyzed under isocratic elusion using 60/40 acetonitrile/water as a mobile phase. The peptides mixture was composed of five peptides, T-T-S, V-A-P-G, AT-I, Isotocine, and Bradykinin, while the mixture of proteins was composed of five proteins, Cytochrome c, RNase A, Lysine, α-Amylase, and Insulin. The number of theoretical plates (N) was calculated using the following equation;
(1)$$\;N=5.54{\left(\frac{t_{r}}{w^{1/2}}\right)}^{2}$$
where w^1/2^ is the peak width at half height, and t_r_ is the retention time of the analyte

## 3. Results and Discussions

### 3.1. Surface Morphology of SP

The SEM images of the bare C18-bound SM particles are shown in Figure 1. The SEM images show that surfaces of the C18-bound SM particles are smoother than the bare SM particles. The attachment of C18 ligands onto the surface of the silanol groups of the silica particles leads to its surface smoothness. The SEM images show that the particles are in the range of 2–3 μm; most of the particles are bigger than 2 μm and less than 3 μm. Moreover, the particles are not totally spherical but a little irregular in shape. These irregular-shaped particles produced a semi-monolithic texture in the packed bed or made some through-flow channels which enabled the smooth flow of the mobile phase and the analyte across the column and gave high permeability to the column.

### 3.2. Particle-Size Analysis of SP

The PSD of the bare SM particles and the C18-bound SM particles are shown in Figure 2. The volume-based PSD of the bare and C18-bound SM particles of this batch and that of our previous study Ref. [33] are shown in Table 1. The d(0.5) of the bare SM particles and the C18-bound SM particles of the current batch was 2.02 and 3.24, respectively, while the PSD of the bare SM particles and the C18-bound SM particles of the previous study was 1.46 and 3.36, respectively. The PSD of this batch is narrower as compared to our previous studies, and most of the particles are in the range of 2–3 μm [26,27].

### 3.3. Pore-Size Distribution of Stationary Phase

The pore-size analysis results of the bare and the C18-bound SM particles are shown in Figure 3. The results of the bare and C18-bound SM particles of this study and our previous study [27] are given in Table 2. The results in Table 2 show that the pore size of the bare SM particles is 310 Å, while that of the C18-bound SM is 241 Å. It means that the pore size of the SM particles reduced after the C18 attachment, which was also observed in our previous study, where the pore size was reduced from 295 Å to 232 Å, as shown in Table 2. Similarly, the pore volumes of the SM particles and the C18-bound SM particles are 0.67 to 0.58 cm^3^/g, respectively, which show that the pore volume of the SM particles reduced after C18 modification [27]. The surface area of the bare SM particles and the C18-bound SM particles is 116 m^2^/g and 105 m^2^/g, respectively, which is in close agreement with our previous work, where the specific surface areas of the bare SM particles and the C18-bound SM particles were 124 m^2^/g and 111 m^2^/g, respectively. The same trend of decrease in specific surface area after chemical modification was observed in this study.

### 3.4. Chromatographic Performance of C18-Bound SM Column

#### 3.4.1. Separation of Peptides

The separation of the five peptides mixture of T-T-S, V-A-P-G, angiotensin-I, isotocin, and bradykinin on the C18-bound SM column (1.8 mm × 100 mm) using 70/30 ACN/water with 0.1% TFA is shown in Figure 4. At the optimum flow rate of 100 µL/min, the number of theoretical plates was 400,000 plates/m, and all five peptides were eluted in 1.5 min, as shown in Figure 4B. For fast analysis, the flow rate was increased to 200 µL/min, and the retention time of the peptides decreased to 1 min (Figure 4A. The chromatogram in Figure 4B shows that all five peptides were eluted within 1 min using the same mobile phase (70/30 acetonitrile/water with 0.1% TFA).

The theoretical plates (N) of peptides separated on the C18-bound SM column (1.8 mm × 100 mm) are given in Table 3. The number of theoretical plates (N) of the peptides for the C18-bound SM column (1.8 mm × 100 mm) is higher than those of the previous study [24], which were 157,000/m and 195,000/m for a 1.8 × 300 mm and a 1.8 × 150 mm column, respectively.

To check the reproducibility of the C18-bound SM column (1.8 mm × 100 mm), three columns of the same dimension were packed and their performances were checked for peptide separation using the same mobile phase and flow rate (100 µL/min). The numbers of the theoretical plates were measured for each column, and the reproducibility data are given in Table 3. Similarly, each column was evaluated for several days, and the reproducibility N and the retention time of the peptides were checked. The day-to-day reproducibility data of the C18-bound SM column (1.8 mm × 100 mm) are shown in Table 3. The results show that the performance of the C18-bound SM column (1.8 mm × 100 mm) is reproducible, and there was very little efficiency lost when the column was operated for a long time (2–3 months). The average relative standard deviations (RSD%) for the batch and time reproducibility are 0.14% and 0.18%, respectively (Table 3).

#### 3.4.2. Separation of Proteins

The C18-bound SM column (1.8 mm × 100 mm) was also checked for the separation of the proteins mixture, composed of Cytochrome c, RNase A, Lysine, α-Amylase, and Insulin. The chromatogram in Figure 5 shows the separation of the proteins mixture on the column (1.8 mm × 100 mm) at optimum conditions: flow rate (100 µL/min) and mobile phase composition (70/30 ACN/ water with 0.1% TFA). Plate counts as high as 235,000 plates/m were achieved for the proteins mixture at the optimum elusion condition. The separation of large proteins is very challenging and achieving high separation efficiency for proteins in HPLC is a big accomplishment. Such high efficiency has not been reported in the literature for protein separation in HPLC. The number of theoretical plates and the reproducibility in N for the proteins is given in Table 4. The *n* values obtained for the protein separation on the C18-bound SM column (1.8 mm × 100 mm) are reproducible, as shown in Table 4. The results indicated that this study is a good contribution to the HPLC column development for peptide and protein separation.

#### 3.4.3. Limit of Detection (DL) and Limit of Quantification (QL) for Analytes

The limit of detection (DL) and the limit of quantification (QL) for each analyte were determined by developing a linear regression curve with relatively very low concentrations of the target compound. The LD and LQ for the selected analytes were calculated from the calibration curve of low concentrations of target analyte, according to Equation (2):(2)$$\mathrm{DL}=3\;\left(\frac{\mathrm{Syx}}{b}\right)$$
where *b* = slope and Syx = standard error of the calibration curve.

The QL values were calculated from the DL values for each analyte, which are given in Table 5. The results show that the average values of the QL for the peptides and proteins were 0.55 ng and 0.48 ng, respectively, using short C18 column (1.8 mm × 100 mm) UV (at 214 nm). The injection volume and the solution concentration were 5 µL and 3.0 × 10^−2^ µg/mL, respectively, at the UV detection (214 nm).

## 4. Conclusions

A reversed phase (RP) SP was prepared by the C18 modification of porous silica monolith particles with chlorodimethyl octadecyl silane. Narrow-bore columns were packed with this stationary phase (C18-bound SM particles) and evaluated for the separation of peptide and protein HPLC. It was observed that the narrow-bore C18-bound SM column (1.8 × 100 mm) successfully separated the peptides and proteins with a very high separation efficiency. For fast analysis, the peptides mixture was analyzed at a high flow rate (200 µL/min), and all five peptides were separated within 1 min with good resolution and efficiency. The second main advantage of the current C18-bound SM column is low column back pressure at a higher flow rate. This lower column back pressure is due to the monolithic texture developed when C18-bound SM particles were packed in the column. The proteins mixture was also separated on the C18-bound SM column with excellent separation efficiency and good resolution. In the literature, there are several articles on the fast analysis of small molecules, such as alkyl benzene, etc., but the separation efficiency and resolution are very poor when compared to the present study. Moreover, the fast analysis of peptides and proteins with such a high efficiency and good resolution has not been reported so far. So, the current study has made room for the development of low-cost, high-efficiency disposable HPLC columns for pharmaceutical companies and research laboratories for the separation of biomolecules.

## Figures and Tables

**Figure 1 polymers-14-02576-f001:**
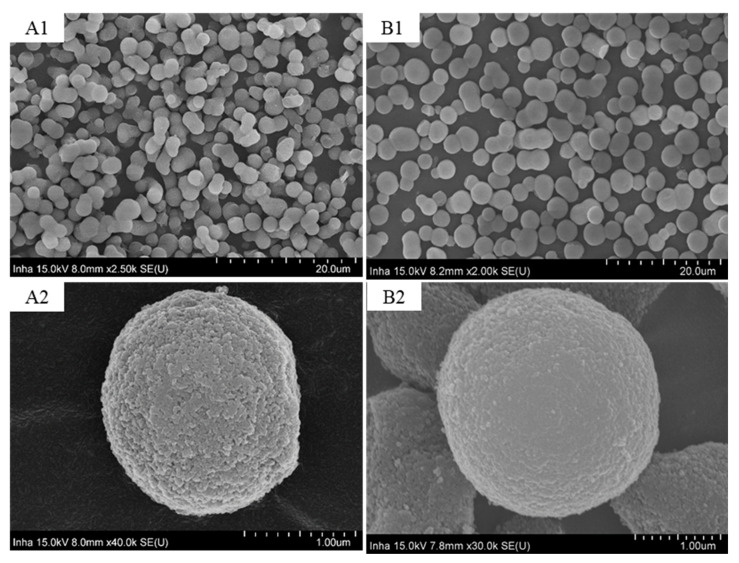
FE-SEM images of bare SM particles (**A1**,**A2**) and C18-bound SM particles (**B1**,**B2**).

**Figure 2 polymers-14-02576-f002:**
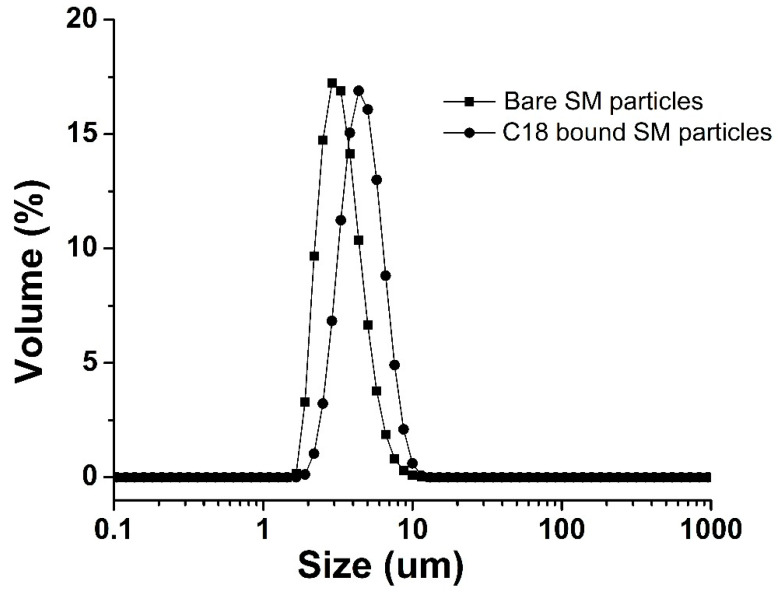
PSD of bare and C18-bound SM particles.

**Figure 3 polymers-14-02576-f003:**
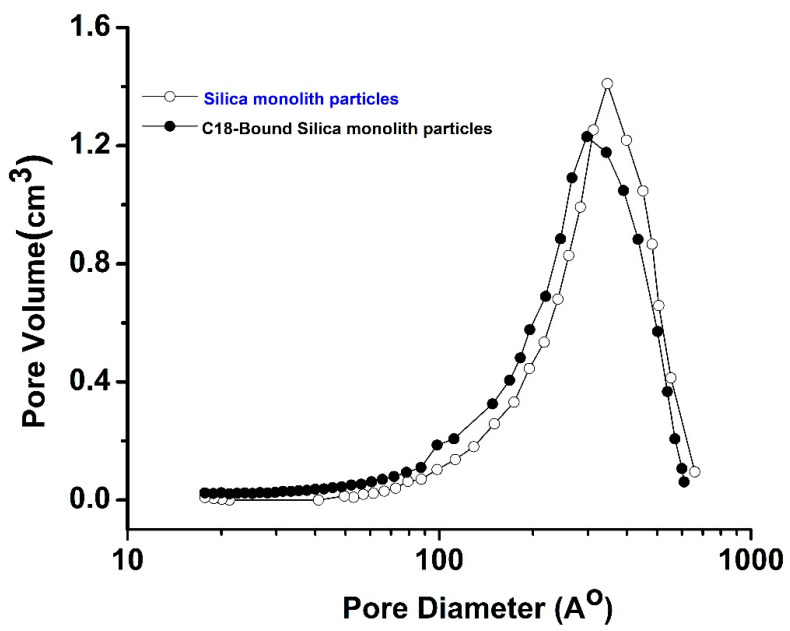
Pore-size distribution of bare and C18-bound SM particles.

**Figure 4 polymers-14-02576-f004:**
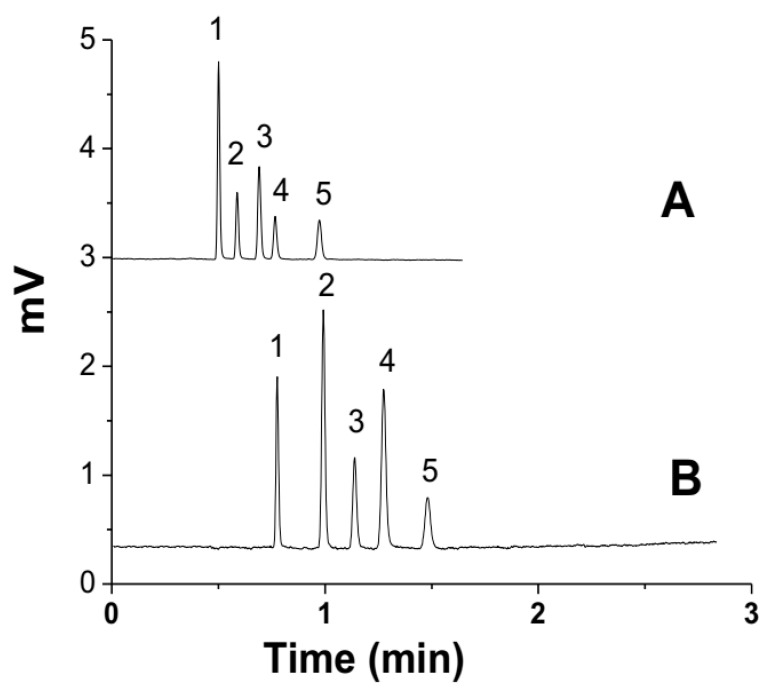
Separation of peptides mixture on C18-bound SM column (1.8 mm × 100 mm) at normal flow rate (100 µL/min) (**B**) and high flow rate (200 µL/min) (**A**); mobile phase: 70/30 ACN/ water with 0.1% TFA. Analytes: 1 (T-T-S), 2 (V-A-P-G), 3 (angiotensin-I), 4 (isotocin), and 5 (bradykinin).

**Figure 5 polymers-14-02576-f005:**
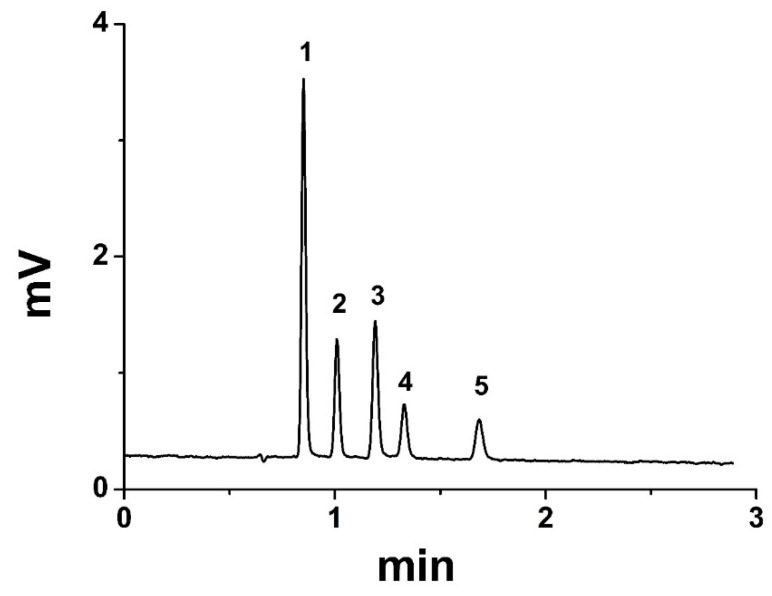
Separation of proteins on C18-bound SM column (1.8 mm × 100 mm); flow rate (100 µL/min), mobile phase 70/30 ACN/water with 0.1% TFA. Analytes: 1 Cytochrome c; 2 RNase A; 3 Lysine; 4 α-Amylase; and 5 Insulin.

**Table 1 polymers-14-02576-t001:** Particle-size distributions of bare and C18-modified silica monolith particles.

	Bare Silica Monolith Particles			C18-Bound Silica Monolith Particles		
	d (0.1)	d (0.5)	d (0.9)	d (0.1)	d (0.5)	d (0.9)
Previous study [33]	0.96	1.46	3.01	1.65	3.36	6.27
Current study	1.23	2.02	5.22	2.25	3.24	6.52

**Table 2 polymers-14-02576-t002:** The pore size, pore volume, and surface area analysis of current and previous SP.

Pore Size (Å)	Bare SM Particles		C18-Bound SM Particles	
	[26]	Current Study	[26]	Current Study
Pore size (Å)	295	310	232	241
Pore volume (cm^3^/g)	0.73	0.67	0.61	0.58
Surface area (m^2^/g)	124	116	111	105

**Table 3 polymers-14-02576-t003:** Batch and time reproducibility of C18-bound SM column (1.8 mm × 100 mm) for peptide separation.

Analyte	Batch Reproducibility		Time Reproducibility	
	*n* Values	%RSD	*n* Values	%RSD
Thr-Tyr-Ser	40,600	0.05	40,200	0.11
Val-Ala-Pro-Gly	40,400	0.10	40,300	0.12
Angiotensin-I	39,500	0.11	39,200	0.18
Isotocin	39,300	0.20	39,000	0.23
Bradykinin	38,200	0.22	38,700	0.28
Average	39,600	0.14	39,400	0.18

**Table 4 polymers-14-02576-t004:** Batch and time reproducibility of C18-bound SM column (1.8 mm × 100 mm) for protein separation.

Analyte	Column-to-Column Reproducibility		Day-to-Day Reproducibility	
	*n* Values	%RSD	*n* Values	%RSD
Cytochrome c	23,300	0.08	23,700	0.15
RNase A	23,200	0.12	23,500	0.18
Lysine	22,500	0.14	22,900	0.28
α-Amylase	21,300	0.29	22,300	0.37
Insulin	21,100	0.32	21,800	0.42
Average	22,200	0.19	22,800	0.28

**Table 5 polymers-14-02576-t005:** DL and QL values determined for peptides and proteins using HPLC-UV detection at 214 nm.

Analyte	DL (ng)	QL (ng)
T-T-S	0.57	1.71
V-A-P-G	0.55	1.65
AT-I	0.51	1.53
Isotocine	0.58	1.74
Bradykinin	0.53	1.59
Cytochrome c	0.49	1.47
RNase A	0.47	1.41
Lysine	0.52	1.56
α-Amylase	0.48	1.44
Insulin	0.45	1.35

## Data Availability

Not applicable.

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
