# Peer review of "Development of Narrow-Bore C18 Column for Fast Separation of Peptides and Proteins in High-Performance Liquid Chromatography"

_polymers, 2022, doi:10.3390/polym14132576_

Round 1
Reviewer 1 Report
The topic is interesting as preparing new chromatographic columns is always necessary. I have some suggestions to improve the manuscript:
1. Add spaces before units such as mm, g an ºC
2. Write Chemicals and Apparatus
3. Add CAS numbers to chemicals
4. Add type of grade of solvents
5. Write compounds such as chlorotrimethylsilane and chlorodimethyloctadecylsilane with lower case (Chemicals & Apparatus) part
6. In the experimental part there is a lack of chromatographic separation part where the type of chromatograph and all conditions of analysis should be explained.
7. Analytical parameters of the method for each compound should be added such as limit of detection and limit of quantification
Author Response
Response to reviewer 1 comments
Dear reviewer of the Polymers, Thank you for your valuable comments on our manuscript entitled “Development of narrow bore C18 column for fast separation of peptides & proteins in high performance liquid chromatography (ID: 1766524)". We very much appreciate the careful reading of our manuscript and the valuable suggestions. We have carefully considered the comments and have revised the manuscript accordingly. We appreciate your suggestions and hope that the corrections will be met with approval.
Authors response to reviewer’s comments are appended below.
- Add spaces before units such as mm, g an ºC
Author response: Spaces were added before each unit throughout the manuscript.
- Write Chemicals and Apparatus
Author response: Chemicals and apparatus are written in experimental section
- Add CAS numbers to chemicals
Author response: CAS numbers are added to chemicals
- Add type of grade of solvents
Author response: Solvents grades are added to each solvent
- Write compounds such as chlorotrimethylsilane and chlorodimethyloctadecylsilane with lower case (Chemicals & Apparatus) part
Author response: The name of these chemicals are written in lower case letters.
- In the experimental part there is a lack of chromatographic separation part where the type of chromatograph and all conditions of analysis should be explained.
Author response: The HPLC analysis part is added to experimental part (section 2.5)
- Analytical parameters of the method for each compound should be added such as limit of detection and limit of quantification
- Author response: Limit of detection & quantification was added to each analyte. See section 3.4.3 Limit of detection (DL) and limit of quantification (QL) for analytes in the revised manuscript and the values are given in table 5.
Reviewer 2 Report
On account of the manuscript POLYMERS-1766524, entitled “Development of narrow bore C18 column for fast separation of peptides & proteins in high performance liquid chromatography” by Ashraf Ali et al., the authors evaluated the separation and performance of the silica monolith (SM) particles prepared by sol-gel method using tri-methoxy orthosilicate (TMOS) as silica source, poly-ethylene glycol (PEG) as surfactant and urea as porogen, which was packed in stainless steel columns. The topic is important to conduct better efficiency with good resolution for chromatograph technologies. However, the manuscript needs critical revision before acceptance for publication. Details of my comments are as follows:
The experiment and the results were elaborate, and the authors got interesting results. However, several concerns are present in the present research. Although the authors mentioned the aim of this study, new aspect or view point of this research was not clear. The methodology and the results were almost the same as those already published in the previous researches, and becoming generally well-known issues today. Therefore, the authors are strongly encouraged to mention the new viewpoints and/or novel aspects which surpass the previous researches globally in the manuscript, and to take these aspects into account to deep the results and discussions with enhanced novelty and better understanding of the results. Otherwise, novelty of the present developed materials for liquid chromatography would not be strengthened. Novelty of research, new approach, clear explanation of methodology, interesting analysis and findings with strong discussion are required for consideration of publication in this Journal. In my opinion, the present manuscript would be more suitable to submit for the Journal such as analytical science and/or applied science.
Author Response
Response to reviewer 2 comments
Dear reviewer of the Polymers, Thank you for your valuable comments on our manuscript submitted to Polymers (ID: 1766524). We have carefully considered your comments and have revised the manuscript accordingly.
On account of the manuscript POLYMERS-1766524, entitled “Development of narrow bore C18 column for fast separation of peptides & proteins in high performance liquid chromatography” by Ashraf Ali et al., the authors evaluated the separation and performance of the silica monolith (SM) particles prepared by sol-gel method using tri-methoxy orthosilicate (TMOS) as silica source, poly-ethylene glycol (PEG) as surfactant and urea as porogen, which was packed in stainless steel columns. The topic is important to conduct better efficiency with good resolution for chromatograph technologies. However, the manuscript needs critical revision before acceptance for publication. Details of my comments are as follows:
The experiment and the results were elaborate, and the authors got interesting results. However, several concerns are present in the present research.
Reviewer comment 1: Although the authors mentioned the aim of this study, new aspect or view point of this research was not clear.
Response: New aspects of current work is the development of short narrow-bore column for fast analysis of peptides and proteins. There are several parameters in chromatography which affect column performance such as stationary phase (particle size, pore size, surface area and bonding chemistry), column packing, which need good skills to pack void free column and method development for separation of targeted analytes (mobile phase selection, mode of separation, flow rate, detection etc.). An improvement in any parameter may change the results of separation significantly. The main purpose of current study was to develop short narrow-bore column for fast analysis of peptides and proteins. In literature, there are several articles on fast analysis but the analytes are small molecules (alkyl benzene etc.) and the separation efficiency and resolution are very poor as compared to the present study.
Reviewer comment 2: The methodology and the results were almost the same as those already published in the previous researches, and becoming generally well-known issues today.
Response: Method of preparation of HPLC stationary phases/columns and mode of operation in LC looks same but there always exist a room for improvement. In current study as mentioned above, the main purpose was to develop short column for peptides and proteins separation which can separate these analytes with high efficiency. The efficiency achieved for peptides and proteins separation under RP-LC mode in the current study is highest and such high efficiency for proteins and peptides fast separation (in shorter time) have not been reported in the literature so far. For instance, there are several articles on fast analysis of small molecules such as alkyl benzene etc. in literature, but the separation efficiency and resolution are very poor as compared to the present study. Moreover, fast analysis of peptides and proteins with such a high efficiency and good resolution have not been reported so far.
Reviewer comment3: Therefore, the authors are strongly encouraged to mention the new viewpoints and/or novel aspects which surpass the previous researches globally in the manuscript, and to take these aspects into account to deep the results and discussions with enhanced novelty and better understanding of the results.
Response: Detail explanation about the novelty of current work has been added in the revised manuscript.
Otherwise, novelty of the present developed materials for liquid chromatography would not be strengthened. Novelty of research, new approach, clear explanation of methodology, interesting analysis and findings with strong discussion are required for consideration of publication in this Journal. In my opinion, the present manuscript would be more suitable to submit for the Journal such as analytical science and/or applied science.
Response: We hope that the addition of required explanation in the revised manuscript will make it suitable for publication in Polymers.
Reviewer 3 Report
Attached please see personlly suggestion.

Author Response
Response to reviewer 3 comments
Reviewer 3 comments: This is interested work. Personally, it can be accepted by Polymers after major revisions:
- The authors should re-organize the presentation to show what the significant updates of current submission is.
Author’s response: Dear reviewer, thank you for your valuable comment and suggestion. We have included the aims of current study and the need of conducting this study as well as novelty in the revised manuscript.
- All the figures should be re-made to reach a high quality. Please see the ACS Journals
Author’s response: The resolution of all figures was increased to 900 dpi. Now it will looks better.
Reviewer 4 Report
The authors claim to have developed a functionalization technique for faster separation in Liquid chromatography.
1. The work looks largely derived and I am not able to find the innovative aspect of this manuscript. The writing is simple and clear.
2. better comparison with bare SM particles is needed
3. Since there seems to be some similar previous work ,it would be prudent to add its findings for comparison of vital parameters.
Author Response
Response to reviewer 4 comments
Reviewer 4 comments
The authors claim to have developed a functionalization technique for faster separation in Liquid chromatography.
- The work looks largely derived and I am not able to find the innovative aspect of this manuscript. The writing is simple and clear.
Response: Dear reviewer, Thank you for your valuable comments, we have defined the research gap and emphasized on the novelty of current work in revised manuscript as suggested by you and other reviewers. We hope in the revised manuscript your concern has been addressed successfully.
- better comparison with bare SM particles is needed
Response: Bare SM particles and ligand bonded SM particles are compared by three ways; 1) SEM analysis, see Figure 1.FE-SEM images of bare SM particles (A1, A2) and C18 bound SM particles (B1, B2). 2) Particle size distribution see Figure 2. PSD of bare and C18 bound SM particles. 3) Pore size distribution, see Fig 3. Pore size distribution of bare and C18 bound SM particles.
- Since there seems to be some similar previous work, it would be prudent to add its findings for comparison of vital parameters.
Response: Particle size, pore size and pore volume of SM particles of current study was compared with previous studies and the results are presented in Table 1 & Table 2. Similarly the number of theoretical plates (N values) of current study and previous studies are compared in section 3.4.1.

Round 2
Reviewer 1 Report
The manuscript has been significantly improved. I don´t have any more suggestions.
Reviewer 2 Report
On account of the manuscript POLYMERS-1766524R1, entitled “Development of narrow bore C18 column for fast separation of peptides & proteins in high performance liquid chromatography” by Ashraf Ali et al., the authors revised the manuscript intensively and appropriately according to the Reviewers comments. After careful consideration, I made a decision that the manuscript is acceptable for publication in its present form.
Reviewer 3 Report
It's ok for publishing now!
Reviewer 4 Report
I recommend the manuscript for publications. they have answered my questions satisfactorily